# Length is a Curse and a Blessing for Document-level Semantics

**Chenghao Xiao**♠  **Yizhi Li**♣  **G Thomas Hudson**♠
**Chenghua Lin**♣  **Noura Al Moubayed**♠
♠ Department of Computer Science, Durham University, UK
♣ Department of Computer Science, The University of Manchester, UK
{chenghao.xiao,g.t.hudson,noura.al-moubayed}@durham.ac.uk
yizhi.li@hotmail.com    chenghua.lin@manchester.ac.uk

## Abstract

In recent years, contrastive learning (CL) has been extensively utilized to recover sentence and document-level encoding capability from pre-trained language models. In this work, we question the length generalizability of CL-based models, i.e., their vulnerability towards length-induced semantic shift. We verify not only that length vulnerability is a significant yet overlooked research gap, but we can devise unsupervised CL methods solely depending on the semantic signal provided by document length. We first derive the theoretical foundations underlying length attacks, showing that elongating a document would intensify the high intra-document similarity that is already brought by CL. Moreover, we found that isotropy promised by CL is highly dependent on the length range of text exposed in training. Inspired by these findings, we introduce a simple yet universal document representation learning framework, **LA(SER)**[3]: length-agnostic self-reference for semantically robust sentence representation learning, achieving state-of-the-art unsupervised performance on the standard information retrieval benchmark. Our code is publicly available.

## 1 Introduction

In recent years, contrastive learning (CL) has become the go-to method to train representation encoder models (Chen et al., 2020; He et al., 2020; Gao et al., 2021; Su et al., 2022). In the field of natural language processing (NLP), the effectiveness of the proposed unsupervised CL methods is typically evaluated on two suites of tasks, namely, semantic textual similarity (STS) (Cer et al., 2017) and information retrieval (IR) (e.g., Thakur et al. (2021)). Surprisingly, a large number of works only validate the usefulness of the learned representations on STS tasks, indicating a strong but widely-adopted assumption that methods optimal for STS could also provide natural transferability to retrieval tasks.

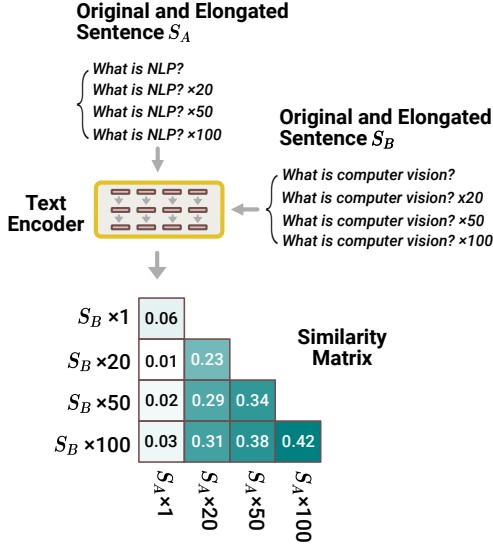

Figure 1: Demonstration of Elongation Attack on Sentence Similarity. The similarity between sentence $S_A$ and $S_B$ incorrectly increases along with elongation, i.e., copy-and-concatenate the original sentence for multiple times, despite the semantics remain unaltered.

Due to the document length misalignment of these two types of tasks, the potential gap in models' capability to produce meaningful representation at different length ranges has been rarely explored (Xiao et al., 2023b). Studies of document length appear to have been stranded in the era where methods are strongly term frequency-based, because of the explicit reflection of document length to sparse embeddings, with little attention given on dense encoders. Length preference for dense retrieval models is observed by Thakur et al. (2021), who show that models trained with dot-product and cosine similarity exhibit different length preferences. However, this phenomenon has not been attributed to the distributional misalignment of length between training and inference domains/tasks, and it remains unknown what abilities of the model are enhanced and diminished when trained with a certain length range.

In this work, we provide an extensive analysis of length generalizability of standard contrastive learning methods. Our findings show that, with default contrastive learning, models' capability to encode document-level semantics largely comes from their coverage of length range in the training.

We first depict through derivation the theoretical underpinnings of the models' vulnerability towards length attacks. Through attacking the documents by the simple copy-and-concatenating elongation operation, we show that the vulnerability comes from the further intensified high intra-document similarity that is already pronounced after contrastive learning. This hinders a stable attention towards the semantic tokens in inference time. Further, we show that, the uniformity/isotropy promised by contrastive learning is heavily length-dependent. That is, models' encoded embeddings are only isotropic on the length range seen in the training, but remain anisotropic otherwise, hindering the same strong expressiveness of the embeddings in the unseen length range.

In the quest to bridge these unideal properties, we propose a simple yet universal framework, **LA(SER)³**: **L**ength-**A**gnostic **SE**lf-**R**eference for **SE**mantically **R**obust **SE**ntence **R**epresentation learning. By providing the simple signal that *"the elongated version of myself 1) should still mean myself, and thus 2) should not become more or less similar to my pairs"*, this framework could not only act as an unsupervised contrastive learning method itself by conducting self-referencing, but could also be combined with any contrastive learning-based text encoding training methods in a plug-and-play fashion, providing strong robustness to length attacks and enhanced encoding ability.

We show that, our method not only improves contrastive text encoders' robustness to length attack without sacrificing their representational power, but also provides them with external semantic signals, leading to state-of-the-art unsupervised performance on the standard information retrieval benchmark.

## 2 Length-based Vulnerability of Contrastive Text Encoders

Length preference of text encoders has been observed in the context of information retrieval (Thakur et al., 2021), showing that contrastive learning-based text encoders trained with dot-product or cosine similarity display opposite length

preferences. Xiao et al. (2023b) further devised "adversarial length attacks" to text encoders, demonstrating that this vulnerability can easily fool text encoders, making them perceive a higher similarity between a text pair by only copying one of them $n$ times and concatenating it to itself.

In this section, we first formalize the problem of length attack, and then analyze the most important pattern (misaligned intra-document similarity) that gives rise to this vulnerability, and take an attention mechanism perspective to derive for the first time the reason why contrastive learning-based text encoders can be attacked.

**Problem Formulation: Simple Length Attack** Given a sentence $S$ with $n$ tokens $\{x_1, x_2, ..., x_n\}$, we artificially construct its elongated version by copying it $m$ times, and concatenating it to itself. For instance, if $m = 2$, this would give us $\widetilde{S} = \{x_1, ..., x_n, x_1, ..., x_n\}$. Loosely speaking, we expect the elongation to be a "semantics-preserved" operation, as repeating a sentence $m$ times does not change the semantics of a sentence in most cases. For instance, in the context of information retrieval, repeating a document $d$ by $m$ time should not make it more similar to a query $q$. In fact, using pure statistical representation such as tf-idf (Sparck Jones, 1972), the original sentence and the elongated version yield exact same representations:

$$\widetilde{S} \triangleq f(S, m) \qquad (1)$$
$$\text{tf-idf}(S) = \text{tf-idf}(\widetilde{S}) \qquad (2)$$

where $f(\cdot)$ denotes the elongation operator, and $m$ is a random integer.

Therefore, no matter according to the semantics-preserved assumption discussed previously, or reference from statistics-based methods (Sparck Jones, 1972), one would hypothesize Transformer-based models to behave the same. Formally, we expect, given a Transformer-based text encoder $g(\cdot)$ to map a document into a document embedding, we could also (**ideally**) get:

$$g(S) = g(\widetilde{S}) \qquad (3)$$

**Observation 1: Transformer-based text encoders perceive different semantics in original texts and elongation-attacked texts.** The central problem is: given a Transformer-based text encoder $g(\cdot)$, it is found empirically that:

$$g(S) \neq g(\widetilde{S}). \qquad (4)$$

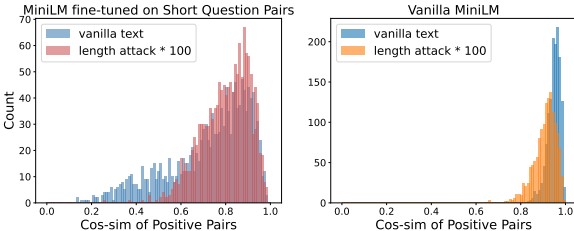

Figure 2: Distribution of positive pair cosine similarity. Left: MiniLM finetuned on only short document pairs with contrastive loss displays a favor towards attacked documents (longer documents). Right: the vanilla model displays an opposite behavior.

We verify this phenomenon with Proof of Concept Experiment 1 (Figures 1, 2), showing that Transformer-based encoders perceive different semantics before and after elongation attacks.

**Proof of Concept Experiment 1** To validate Observation 1, we fine-tune a vanilla MiniLM (Wang et al., 2020) with the standard infoNCE loss (Oord et al., 2018) with in-batch negatives, on the Quora duplicate question pair dataset (QQP). Notably, the dataset is composed of questions, and thus its length coverage is limited (average token length = 13.9, with 98.5% under 30 tokens).

With the fine-tuned model, we first construct two extreme cases: one with a false positive pair ("*what is NLP?*" v.s., "*what is computer vision?*"), one with a positive pair ("*what is natural language processing?*" v.s., "*what is computational linguistics?*"). We compute cosine similarity between mean-pooled embeddings of the original pairs, and between the embeddings attained after conducting an elongation attack with $m = 100$ (Eq. 1).

We found surprisingly that, while "*what is NLP?*" and "*what is computer vision?*" have 0.06 cosine similarity, their attacked versions achieve 0.42 cosine similarity - successfully attacked (cf. Figure 1). And the same between "*what is natural language processing?*" and "*what is computational linguistics?*" goes from 0.50 to 0.63 - similarity pattern augmented.

On a larger scale, we then construct an inference set with all the document pairs from Semantic Textual Similarity benchmark (STS-b) (Cer et al., 2017). We conduct an elongation attack on all sentences with $m = 100$ (Eq. 1). The distributions of document pair cosine similarity are plotted in Figure 2. For the fine-tuned MiniLM (Figure 2, left), it is clearly shown that, the model perceives in general a higher cosine similarity between documents

after elongation attacks, greatly increasing the perceived similarity, even for pairs that are not positive pairs. This phenomenon indicates a built-in vulnerability in contrastive text encoders, hindering their robustness for document encoding. For reference, we also plot out the same set of results on the vanilla MiniLM (Figure 2, right), demonstrating an opposite behavior, which will be further discussed in Proof of Concept Experiment 2.

**Observation 2: Intra-document token interactions experience a pattern shift after elongation attacks.** Taking an intra-document similarity perspective (Ethayarajh, 2019), we can observe that, tokens in the elongated version of same text, do not interact with one another as they did in the original text (see Proof of Concept Experiment 2). Formally, given tokens in $S$ providing an intra-document similarity of $sim$, and tokens in the elongated version $\widetilde{S}$ providing $\tilde{sim}$, we will show that $sim \neq \tilde{sim}$. This pattern severely presents in models that have been finetuned with a contrastive loss, while is not pronounced in their corresponding vanilla models (PoC Experiment 2, Figure 3).

A significant increase on intra-document similarity of contrastive learning-based models is observed by Xiao et al. (2023a), opposite to their vanilla pre-trained checkpoints (Ethayarajh, 2019). It is further observed that, after contrastive learning, semantic tokens (such as topical words) become *dominant* in deciding the embedding of a sentence, while embeddings of functional tokens (such as stop-words) follow wherever these semantic tokens travel in the embedding space. This was formalized as the "entourage effect" (Xiao et al., 2023a). Taking into account this conclusion, we further derive from the perspective of attention mechanism, the reason why conducting elongation attacks would further intensify the observed high intra-document similarity.

The attention that any token $x_i$ in the sentence $S$ gives to the dominant tokens can be expressed as:

$$\text{Attention}(\underset{i \in S}{x_i} \rightarrow x_{\text{dominant}}) = \frac{e^{q_i k_{dominant}^T / \sqrt{d_k}}}{\sum_n e^{q_i k_n^T / \sqrt{d_k}}},$$
(5)

where $q_i$ is the query vector produced by $x_i$, $k_{dominant}^T$ is the transpose of the key vector produced by $x_{dominant}$, and $k_n^T$ is the transpose of the key vector produced by every token $x_n$. We omit the $V$ matrix in the attention formula for simplicity.

After elongating the sentence $m$ times with the copy-and-concat operation, the attention distribution across tokens shifts, taking into consideration that the default prefix `[cls]` token is not elongated. Therefore, in inference time, `[cls]` tokens share less attention than in the original sentence.

To simplify the following derivations, we further impose the assumption that positional embeddings contribute little to representations, which loosely hold empirically in the context of contrastive learning (Yuksekgonul et al., 2023). In Section 6, we conduct an extra group of experiment to present the validity of this imposed assumption by showing the positional invariance of models after CL.

With this in mind, after elongation, the same token in different positions would get the same attention, because they have the same token embedding without positional embeddings added. Therefore:

$$\widetilde{\text{Attention}}(\underset{i \in \widetilde{S}}{x_i \to x_{\text{dominant}}})$$
$$= \frac{m e^{q_i k_{dominant}^T / \sqrt{d_k}}}{m \sum_n e^{q_i k_n^T / \sqrt{d_k}} - (m-1) e^{q_i k_{[cls]}^T / \sqrt{d_k}}}$$
$$= \frac{e^{q_i k_{dominant}^T / \sqrt{d_k}}}{\sum_n e^{q_i k_n^T / \sqrt{d_k}} - \frac{m-1}{m} e^{q_i k_{[cls]}^T / \sqrt{d_k}}} \quad (6)$$
$$> \text{Attention}(\underset{i \in S}{x_i \to x_{\text{dominant}}})$$

Based on Eq. 6, we can see that attentions towards dominant tokens would increase after document elongation attack. However, we can also derive that the same applies to non-dominant tokens:

$$\widetilde{\text{Attention}}(\underset{i \in \widetilde{S}}{x_i \to x_{\text{non-dominant}}})$$
$$> \text{Attention}(\underset{i \in S}{x_i \to x_{\text{non-dominant}}})$$

In fact, every unique token except `[cls]` would experience an attention gain. Therefore, we have to prove that, the attention gain $G_d$ of dominant tokens (denoted as $x_d$) outweighs the attention gain $G_r$ of non-dominant (regular, denoted as $x_r$) tokens. To this end, we define:

$$G_d$$
$$\triangleq \widetilde{\text{Attention}}(\underset{i \in \widetilde{S}}{x_i \to x_{\text{d}}}) - \text{Attention}(\underset{i \in S}{x_i \to x_{\text{d}}})$$
$$(7)$$

$$G_r$$
$$\triangleq \widetilde{\text{Attention}}(\underset{i \in \widetilde{S}}{x_i \to x_{\text{r}}}) - \text{Attention}(\underset{i \in S}{x_i \to x_{\text{r}}})$$
$$(8)$$

Let $e^{q_i k_{dominant}^T / \sqrt{d_k}}$ be $l_d$, $e^{q_i k_{non\text{-}dominant}^T / \sqrt{d_k}}$ be $l_r$, $e^{q_i k_n^T / \sqrt{d_k}}$ be $l_n$, and $e^{q_i k_{[cls]}^T / \sqrt{d_k}}$ be a $l_c$, we get:

$$G_d$$
$$\triangleq \widetilde{\text{Attention}}(\underset{i \in \widetilde{S}}{x_i \to x_{\text{d}}}) - \text{Attention}(\underset{i \in S}{x_i \to x_{\text{d}}})$$
$$= \frac{l_d}{\sum_n l_n - \frac{m-1}{m} l_c} - \frac{l_d}{\sum_n l_n} = \frac{l_d \frac{m-1}{m} l_c}{\sum_n l_n (\sum_n l_n - \frac{m-1}{m} l_c)}$$
$$(9)$$

Similarly, we get:

$$G_r = \frac{l_r \frac{m-1}{m} l_c}{\sum_n l_n (\sum_n l_n - \frac{m-1}{m} l_c)} \quad (10)$$

Also note that $l_d > l_r$: that's why they are called "dominating tokens" in the first place (Xiao et al., 2023a). Therefore, we prove that $G_d > G_r$.

As a result, with elongation operation, every token is going to assign even more attention to the embeddings of the dominating tokens. And this effect propagates throughout layers, intensifying the high intra-document similarity ("entourage effect") found in (Xiao et al., 2023a).

**Proof of Concept Experiment 2** With the derivations, we conduct PoC Experiment 2, aiming to demonstrate that intra-document similarity experiences a pattern shift after elongation attack, intensifying the "entourage effect", for contrastive fine-tuned models.

Taking the same fine-tuned MiniLM checkpoint from PoC Experiment 1, we compute the intra-document similarity of all the model outputs on STS-b. For each document, we first compute its document embedding by mean-pooling, then compute the average cosine similarity between each token embedding and the document embedding.[1] The results are shown in Figure 3. After elongation attacks, we can see an increase in the already high

---

[1]Notably, we further adjust these scores by the model's anisotropy estimation (average pair-wise similarity of random sampled tokens), because of the representation degeneration problem (Gao et al., 2019; Ethayarajh, 2019).

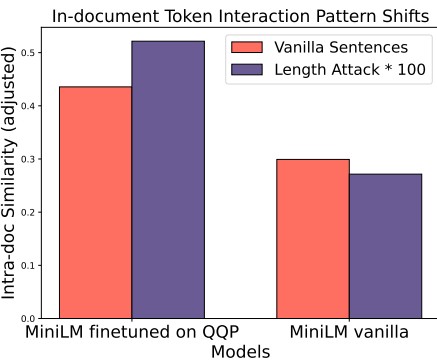

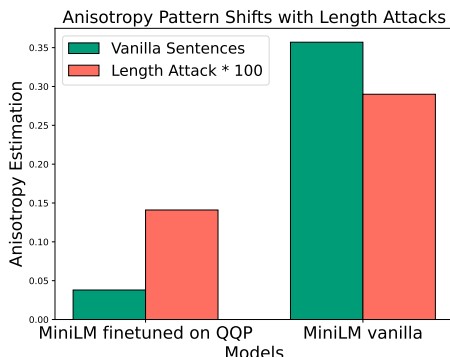

Figure 3: In-document Token Interactions experience a pattern shift before and after contrastive fine-tuning: Using the vanilla model, tokens in the elongated version of a document become less like one another than in the original un-attacked text; after contrastive fine-tuning, tokens in the attacked text look more alike to one another. This empirically validates our math derivation. Notably, measurements of both models have been adjusted by their anisotropy estimation (displayed value = avg. intra similarity - estimated anisotropy value).

Figure 4: Isotropy Pattern Shifts. Albeit contrastive learning has an isotropy promise, we question this by showing the model is only isotropic in its trained length range, remaining anisotropic otherwise (shown by increased anisotropy after length attacks).

intra-document similarity, meaning that all other tokens converge even further towards the tokens that dominate the document-level semantics.

When using the vanilla MiniLM checkpoint, the intra-document similarity pattern is again reversed. This opposite pattern is well-aligned with the findings of Ethayarajh (2019) and Xiao et al. (2023a): Because in vanilla language models, the intra-document similarity generally becomes lower in the last few layers, while after contrastive learning, models show a drastic increase of intra-document similarity in the last few layers. Also, our derivations conclude that: if the intra-document similarity shows an accumulated increase in the last few layers, this increase will be intensified after elongation; and less affected otherwise.

Complementing the intensified intra-document similarity, we also display an isotropy misalignment before and after elongation attacks in Figure 4. With the well-known representation degeneration or anisotropy problems in vanilla pre-trained models (Figure 4, right, green, Gao et al. (2019); Ethayarajh (2019)), it has been previously shown that after contrastive learning, a model's encoded embeddings will be promised with a more isotropic geometry (Figure 4, left, green, Wang and Isola (2020); Gao et al. (2021); Xiao et al. (2023a)). However, in this work, we question this general conclusion by showing that the promised isotropy is strongly length-dependent. After elongation, the

embeddings produced by the fine-tuned checkpoint start becoming anisotropic (Figure 4, left, pink). This indicates that, if a model has only been trained on short documents with contrastive loss, only the short length range is promised with isotropy.

On the other hand, elongation attacks seem to be able to help vanilla pre-trained models to escape from anisotropy, interestingly (Figure 4, right, pink). However, the latter is not the key focus of this work.

## 3   Method: LA(SER)$^3$

After examining the two fundamental reasons underlying the built-in vulnerability brought by standard contrastive learning, the formulation of our method emerges as an intuitive outcome. Naturally, we explore the possibility of using only *length* as the semantic signal to conduct contrastive sentence representation learning, and propose **LA(SER)$^3$**: Length-Agnostic Self-Reference for Semantically Robust Sentence Representation Learning. LA(SER)$^3$ builds upon the semantics-preserved assumption that *"the elongated version of myself 1) should still mean myself, and thus 2) should not become more or less similar to my pairs"*. **LA(SER)$^3$** leverages elongation augmentation during the unsupervised constrastive learning to improve 1) the robustness of in-document interaction pattern in inference time; 2) the isotropy of larger length range. We propose two versions of reference methods, for different format availability of sentences in target training sets.

| Models → | SimCSE† | | ESimCSE† | | DiffCSE† | | InfoCSE†♠ | | LA(SER)³ (Ours) | |
|---|---|---|---|---|---|---|---|---|---|---|
| Test Dataset ↓ | base | large | base | large | base | large | base | large | base-64 | base-128 |
| trec-covid | 0.2750 | 0.2264 | 0.2291 | 0.2829 | 0.2368 | 0.2291 | **0.3937** | 0.3166 | 0.2728 | 0.3463 |
| nfcorpus | 0.1048 | 0.1356 | 0.1149 | 0.1483 | 0.1204 | 0.1470 | 0.1358 | 0.1576 | 0.1652 | **0.1919** |
| nq | 0.1628 | 0.1671 | 0.0935 | 0.1705 | 0.1188 | 0.1556 | **0.2023** | 0.1790 | 0.1556 | 0.1354 |
| fiqa | 0.0985 | 0.0975 | 0.0731 | **0.1117** | 0.0924 | 0.1027 | 0.0991 | 0.1000 | 0.1057 | 0.1090 |
| arguana | 0.2796 | 0.2078 | 0.3376 | 0.2604 | 0.2500 | 0.2572 | 0.3244 | 0.4133 | 0.4182 | **0.4227** |
| webis-touche2020 | **0.1342** | 0.0878 | 0.0786 | 0.1057 | 0.0912 | 0.0781 | 0.0935 | 0.0920 | 0.1105 | 0.1167 |
| quora | 0.7375 | 0.7511 | 0.7411 | 0.7615 | 0.7491 | 0.7471 | 0.8241 | **0.8268** | 0.7859 | 0.7741 |
| cqadupstack | 0.1349 | 0.1082 | 0.1276 | 0.1196 | 0.1197 | 0.1160 | **0.2097** | 0.1881 | 0.1687 | 0.1691 |
| dbpedia-entity | 0.1662 | 0.1495 | 0.1260 | 0.1650 | 0.1537 | 0.1571 | **0.2101** | 0.1838 | 0.1645 | 0.1663 |
| scidocs | 0.0611 | 0.0688 | 0.0657 | 0.0796 | 0.0673 | 0.0699 | 0.0837 | **0.0859** | 0.0764 | **0.0859** |
| climate-fever | **0.1420** | 0.1065 | 0.0796 | 0.1302 | 0.1019 | 0.1087 | 0.0937 | 0.0840 | 0.1311 | 0.1197 |
| scifact | 0.2492 | 0.2541 | 0.3013 | 0.2875 | 0.2666 | 0.2811 | 0.3269 | 0.3801 | 0.3960 | **0.4317** |
| hotpotqa | 0.2382 | 0.1896 | 0.1213 | 0.1970 | 0.1730 | 0.2068 | **0.3177** | 0.2781 | 0.2827 | 0.2937 |
| fever | **0.2916** | 0.1776 | 0.0756 | 0.1689 | 0.1416 | 0.1849 | 0.1978 | 0.1252 | 0.2388 | 0.2691 |
| average | 0.2197 | 0.1948 | 0.1832 | 0.2135 | 0.1916 | 0.2030 | 0.2509 | 0.2436 | 0.2480 | **0.2594** |

Table 1: Unsupervised BERT nDCG@10 performances on BEIR information retrieval benchmark. †: Results are from Wu et al. (2022a). ♠: Unfair comparison. Notably, InfoCSE benefits from the pre-training of an auxiliary network, while the rest of the baselines and our method fully rely on unsupervised contrastive fine-tuning on the same training^wiki setting as described in §4. Note that with a batch size of 64, our method already outperforms all baselines to a large margin except InfoCSE. Since we train with a max sequence length of 256 (all baselines are either 32 or 64), we find that training with a larger batch size (128) further stabilizes our training, achieving state-of-the-art results. Further, we achieve state-of-the-art with only a BERT_base.

**Self-reference** In LA(SER)³_self-ref setting, we take a sentence from the input as an anchor for each training input, and construct its positive pair by elongating the sentence to be $m$ times longer.

**Intra-reference** LA(SER)³_intra-ref conducts intra-reference within the document. The two components of a positive pair are constructed from different spans of the same document. Since we are only to validate effectiveness of LA(SER)³_intra-ref, we implement this in the simple mutually-excluded span setting. In other words, the LA(SER)³_intra-ref variant takes a sentence (either the first or a random sentence) from the text as an anchor, uses the rest of the text in the input as its positive pair, and elongates the anchor sentence $m$ times as the augmented anchor.

For both versions, we use the standard infoNCE loss (Oord et al., 2018) with in-batch negatives as the contrastive loss.

## 4 Experiments

**Training datasets** We conduct our experiments on two training dataset settings: 1) training^wiki uses 1M sentences sampled from Wikipedia, in line with previous works on contrastive sentence representation learning (Gao et al., 2021; Wu et al., 2022a,b); 2) training^msmarco uses MSMARCO

(Nguyen et al., 2016), which is equivalent to in-domain-only setting of the BEIR information retrieval benchmark (Thakur et al., 2021).

**Evaluation datasets** The trained models are mainly evaluated on the BEIR benchmark (Thakur et al., 2021), which comprises 18 datasets on 9 tasks (fact checking, duplicate question retrieval, argument retrieval, news retrieval, question-answering, tweet retrieval, bio-medical retrieval and entity retrieval). We evaluate on the 14 public zero-shot datasets from BEIR (BEIR-14). And we use STS-b (Cer et al., 2017) only as the auxiliary experiment.

The reasons why we do not follow the *de facto* practice, which mainly focuses on cherry-picking the best training setting that provides optimal performance on STS-b are as follows: Firstly, performances on STS-b do not display strong correlations with downstream tasks (Reimers et al., 2016). In fact, document-level encoders that provide strong representational abilities do not necessarily provide strong performance on STS-b (Wang et al., 2021). Furthermore, recent works have already attributed the inferior predictive power of STS-b performance on downstream task performances to its narrow length range coverage (Abe et al., 2022). Therefore, we believe a strong sentence and document-level representation encoder should be evaluated

| Training Setting→ | | Trained on wiki Self-reference | | | Trained on MSMARCO Self-reference | | | Trained on MSMARCO Intra-reference | | |
|---|---|---|---|---|---|---|---|---|---|---|
| | Models → Test dataset ↓ | SimCSE | LA(SER)$^3$ | Perf. Gain | SimCSE | LA(SER)$^3$ | Perf. Gain | COCO-DR (PT-unsup) | LA(SER)$^3$ | Perf. Gain |
| zero-shot setting | **trec-covid** | 0.1473 | 0.2129 | **44.52%** | 0.1467 | 0.1646 | **12.22%** | 0.2597 | 0.2511 | **-3.33%** |
| | **nfcorpus** | 0.0764 | 0.1265 | **65.54%** | 0.0796 | 0.0933 | **17.31%** | 0.1853 | 0.1508 | **-18.62%** |
| | **nq** | 0.0370 | 0.0836 | **125.88%** | 0.0302 | 0.0391 | **29.55%** | 0.0268 | 0.0405 | **51.10%** |
| | **fiqa** | 0.0288 | 0.0590 | **104.94%** | 0.0260 | 0.0435 | **67.36%** | 0.0821 | 0.1030 | **25.48%** |
| | **arguana** | 0.2277 | 0.3130 | **37.48%** | 0.2081 | 0.1961 | **-5.74%** | 0.3441 | 0.3834 | **11.42%** |
| | **webis-touche2020** | 0.0289 | 0.0483 | **66.99%** | 0.0177 | 0.0296 | **67.71%** | 0.0736 | 0.0896 | **21.73%** |
| | **quora** | 0.6743 | 0.7095 | **5.22%** | 0.6527 | 0.6515 | **-0.19%** | 0.7976 | 0.7911 | **-0.82%** |
| | **cqadupstack** | 0.0889 | 0.1279 | **43.90%** | 0.0864 | 0.1105 | **27.95%** | 0.1380 | 0.1560 | **13.06%** |
| | **dbpedia-entity** | 0.0837 | 0.1138 | **36.04%** | 0.0541 | 0.0558 | **3.03%** | 0.0924 | 0.0825 | **-10.76%** |
| | **scidocs** | 0.0259 | 0.0516 | **99.54%** | 0.0178 | 0.0309 | **73.19%** | 0.0305 | 0.0492 | **61.56%** |
| | **climate-fever** | 0.0127 | 0.0789 | **522.24%** | 0.0136 | 0.0198 | **45.11%** | 0.0652 | 0.1108 | **69.84%** |
| | **scifact** | 0.2174 | 0.3525 | **62.12%** | 0.2330 | 0.2276 | **-2.32%** | 0.4056 | 0.4076 | **0.49%** |
| | **hotpotqa** | 0.0829 | 0.1646 | **98.56%** | 0.0560 | 0.0750 | **34.07%** | 0.0383 | 0.0539 | **40.85%** |
| | **fever** | 0.0363 | 0.1001 | **175.88%** | 0.0263 | 0.0340 | **29.41%** | 0.1421 | 0.2524 | **77.60%** |
| | **average** | 0.1263 | 0.1816 | **43.78%** | 0.1177 | 0.1265 | **7.48%** | 0.1915 | **0.2087** | **8.97%** |

Table 2: Unsupervised Performance Trained with MiniLM-L6 Model. For self-reference settings, we compare with SimCSE (Gao et al., 2021). Notably, LA(SER)$^3{}_{\text{self-ref}}$ can be viewed as a plug-and-play module to SimCSE, as SimCSE takes an input itself as both the anchor and the positive pair, while LA(SER)$^3{}_{\text{self-ref}}$ further elongates this positive pair. For intra-referecne setting, we compare with COCO-DR (Yu et al., 2022). Notabaly, we only experiment with the unsupervised pre-training part of COCO-DR, as LA(SER)$^3{}_{\text{intra-ref}}$ can be viewed as a plug-and-play module to this part. We believe combining with our method for a better unsupervised pretrained checkpoint, the follow-up supervised fine-tuning in COCO-DR can further achieve better results.

beyond semantic textual similarity tasks. However, for completeness, we also provide the results of STS-b in Appendix A.

**Baselines** We compare our methods in two settings, corresponding to the two versions of **LA(SER)$^3$**: 1) Self-Reference. Since we assume using the input itself as its positive pair in this setting, it is natural to compare LA(SER)$^3{}_{\text{self-ref}}$ to the strong baseline SimCSE (Gao et al., 2021). In the `training`$^{\texttt{wiki}}$ setting, we further compare with E-SimCSE, DiffCSE, and InfoCSE (Wu et al., 2022b; Chuang et al., 2022; Wu et al., 2022a). Notably, these four baselines all have public available checkpoints trained on `training`$^{\texttt{wiki}}$.

2) Intra-Reference. The baseline method in this case is: taking a sentence (random or first) from a document as anchor, then use the remaining content of the document as its positive pair. Notably, this baseline is similar to the unsupervised pretraining part of COCO-DR (Yu et al., 2022), except COCO-DR only takes two sentences from a same document, instead of one sentence and the remaining part. Compared to the baseline, LA(SER)$^3{}_{\text{intra-ref}}$ further elongates the anchor sentence. In the result table, we refer to baseline of this settings as COCO-DR$_{\text{PT-unsup}}$.

**Implementation Details** We evaluate the effectiveness of our method with BERT (Devlin et al., 2019) and MiniLM[2] (Wang et al., 2020). To compare to previous works, we first train a LA(SER)$^3{}_{\text{self-ref}}$ on `training`$^{\texttt{wiki}}$ with BERT-base (uncased). We then conduct most of our in-depth experiments with vanilla MiniLM-L6 due to its low computational cost and established state-of-the-art potential after contrastive fine-tuning.[3]

All experiments are run with 1 epoch, a learning rate of 3e-5, a temperature $\tau$ of 0.05, a max sequence length of 256, and a batch size of 64 unless stated otherwise. All experiments are run on Nvidia A100 80G GPUs.

Notably, previous works on contrastive sentence representation learning (Gao et al., 2021; Wu et al., 2022b; Chuang et al., 2022; Wu et al., 2022a) and even some information retrieval works such as (Yu et al., 2022) mostly use a max sequence length of 32 to 128. In order to study the effect of length, we set the max sequence length to 256, at the cost of constrained batch sizes and a bit of computational overhead. More detailed analyses on max sequence length are in ablation analysis (§5).

For the selection of the anchor sentence, we take the first sentence of each document in the main experiment (we will discuss taking a random sen-

[2]We use a 6-layer version by taking every second layer. https://huggingface.co/nreimers/MiniLM-L6-H384-uncased

[3]For instance, sentence-transformers/all-MiniLM-L6-v2 is a SOTA sentence encoder fine-tuned with MiniLM-L6.

tence instead of the first sentence in the ablation analysis in §5.1). For LA(SER)$^3$_self-ref, we elongate the anchor sentence to serve as its positive pair; for LA(SER)$^3$_intra-ref, we take the rest of the document as its positive pair, but then elongate the anchor sentence as the augmented anchor. For the selection of the elongation hyperparameter $m$, we sample a random number for every input depending on its length and the max length of 256. For instance, if a sentence has 10 tokens excluding [cls], we sample a random integer from [1,25], making sure it is not exceeding maximum length; while for a 50-token sentence, we sample from [1, 5]. We will discuss the effect of elongating to twice longer, instead of random-times longer in ablation §5.2.

**Results** The main results are in Tables 1 and 2. Table 1 shows that our method leads to state-of-the-art average results compared to previous public available methods and checkpoints, when training on the same training$^{wiki}$ with BERT.

Our method has the exact same setting (training a vanilla BERT on the same training$^{wiki}$) with the rest of the baselines except InfoCSE, which further benefits from the training of an auxiliary network. Note that with a batch size of 64, our method already outperforms all the baselines to a large margin except InfoCSE. Since we train with a max sequence length of 256 (all baselines are either 32 or 64), we find that training with a larger batch size (128) further stabilizes our training, achieving state-of-the-art results. Moreover, we achieve state-of-the-art with only a BERT_base.

In general, we find that our performance gain is more pronounced when the length range of the dataset is large. On BERT-base experiments, large nDCG@10 performance gain is seen on NFCorpus (doc. avg. length 232.26, SimCSE: 0.1048 -> LA(SER)$^3$: 0.1919), Scifact (doc. avg. length 213.63, SimCSE: 0.2492 -> LA(SER)$^3$: 0.4317), Arguana (doc. avg. length 166.80, SimCSE: 0.2796 -> LA(SER)$^3$: 0.4227). On the other hand, our performance gain is limited when documents are shorter, such DBPedia (avg. length 49.68) and Quora (avg. length 11.44).

Table 2 further analyzes the effect of datasets and LA(SER)$^3$ variants with MiniLM-L6, showing a consistent improvement when used as a plug-and-play module to previous SOTA methods.

We also found that, even though MiniLM-L6 shows great representational power if after supervised contrastive learning with high-quality doc-

ument pairs (see popular Sentence Transformers checkpoint all-MiniLM-L6-v2), its performance largely falls short under unsupervised training settings, which we speculate to be due to that the linguistic knowledge has been more unstable after every second layer of the model is taken (from 12 layers in MiniLM-L12 to 6 layers). Under such setting, LA(SER)$^3$_intra-ref largely outperforms LA(SER)$^3$_self-ref, by providing signals of more lexical differences in document pairs.

## 5 Ablation Analysis

In this section, we ablate two important configurations of LA(SER)$^3$. Firstly, the usage of LA(SER)$^3$ involves deciding which sentence in the document to use as the anchor (§ 5.1). Secondly, how do we maximize the utility of self-referential elongation? Is it more important for the model to know "me * m = me", or is it more important to cover a wider length range (§ 5.2)?

### 5.1 Selecting the Anchor: first or random?

If a document consists of more than one sentence, LA(SER)$^3$ requires deciding which sentence in the document to use as the anchor. We ablate this with both LA(SER)$^3$_self-ref and LA(SER)$^3$_intra-ref on training$^{msmarco}$, because training$^{wiki}$ consists of mostly one-sentence inputs and thus is not able to do intra-ref or random sentence.

| Anchor Sentence | Method | Zero-shot Average | Performance Change |
|---|---|---|---|
| First | SimCSE
LA(SER)$^3$_self-ref | 0.1177
0.1265 | ↑7.48% |
| Random | SimCSE
LA(SER)$^3$_self-ref | 0.1127
0.1013 | ↓10.05% |
| First | COCO-DR_pt-unsup
LA(SER)$^3$_intra-ref | 0.1915
**0.2087** | ↑ 8.97% |
| Random | COCO-DR_pt-unsup
LA(SER)$^3$_intra-ref | 0.1930
0.2033 | ↑ 5.33% |

Table 3: Taking First sentence or Random sentence as the anchor? - ablated with MiniLM-L6 on training$^{msmarco}$.

The results are in Table 3. In general, we observe that taking a random sentence as anchor brings certain noise. This is most corroborated by the performance drop of LA(SER)$^3$_self-ref + random sentence, compared to its SimCSE baseline. However, LA(SER)$^3$_intra-sim + random sentence seems to be able to act robustly against this noise.

We hypothesize that as LA(SER)$^3$ provides augmented semantic signals to contrastive learning, it would be hurt by overly noisy in-batch inputs. By contrast, LA(SER)$^3_{\text{intra-sim}}$ behaves robustly to this noise because the rest of the document apart from the anchor could serve as a stabilizer to the noise.

## 5.2 Importance of Self-referential Elongation

With the validated performance gain produced by the framework, we decompose the inner-workings by looking at the most important component, elongation. A natural question is: is the performance gain only brought by coverage of larger trained length range? Or does it mostly rely on the semantic signal that, "my-longer-self" still means myself?

| Elongation Mode | Max Seq. Length | Zero-shot Average |
|---|---|---|
| None | 256 | 0.1263 |
| Twice | 256 | 0.1523 |
| Random | 64 | 0.1778 |
| | 128 | 0.1764 |
| | 256 | **0.1816** |

Table 4: 1) Elongdating to fixed-times longer or a random time? 2) Do length range coverage matter? - ablated with MiniLM-L6 on training$^{\text{wiki}}$.

Table 4 shows that, elongating to random-times longer outperforms elongating to a fixed two-times longer. We hypothesize that, a fixed augmentation introduces certain overfitting, preventing the models to extrapolate the semantic signal that "elongated me = me". On the other hand, as long as they learn to extrapolate this signal (by * random times), increasing max sequence length provides decreasing marginal benefits.

## 6 Auxiliary Property Analysis

### 6.1 Positional Invariance

Recalling in Observation 2 and PoC experiment 2, we focused on analyzing the effect of elongation attack on intra-sentence similarity, which is already high after CL (Xiao et al., 2023a). Therefore, we have imposed the absence of positional embeddings with the aim to simplify the derivation in proving that, with elongation, dominant tokens receive higher attention gains than regular tokens. Here, we present the validity of this assumption by showing models' greatly reduced sensitivity towards positions after contrastive learning.

We analyze the positional (in)sensitivity of 4 models (MiniLM (Wang et al., 2020) and mpnet (Song et al., 2020) respectively before and after contrastive learning on Sentence Embedding Training Data[4]). Models after contrastive learning are Sentence Transformers (Reimers and Gurevych, 2019) models `all-mpnet-base-v2` and `all-MiniLM-L12-v2`.

We take the sentence pairs from STS-b test set as the inference set, and compute each model's perceived cosine similarity on the sentence pairs (distribution 1). We then randomly shuffle the word orders of all sentence 1s in the sentence pairs, and compute each model's perceived cosine similarity with sentence 2s again (distribution 2).

The divergence of the two distributions for each model can serve as a proxy indicator of the model's sensitivity towards word order, and thus towards positional shift. The lower the divergence, the more insensitive that a model is about positions.

We find that the Jenson Shannon divergence yielded by MiniLM has gone from 0.766 (vanilla) to 0.258 (after contrastive learning). And the same for mpnet goes from 0.819 (vanilla) to 0.302 (after contrastive learning). This finding shows that contrastive learning has largely removed the contribution of positions towards document embeddings, even in the most extreme case (with random shuffled word orders). This has made contrastively-learned models acting more like bag-of-words models, aligning with what was previously found in vision-language models (Yuksekgonul et al., 2023).

Moreover, MiniLM uses absolute positional embeddings while mpnet further applies relative positional embeddings. We believe that the positional insensitivity pattern holds for both models can partly make the pattern and **LA(SER)$^3$**'s utility more universal, especially when document encoders are trained with backbone models that have different positional encoding methods.

## 7 Conclusion

In this work, we questioned the length generalizability of contrastive learning-based text encoders. We observed that, despite their seemingly strong representational power, this ability is strongly vulnerable to length-induced semantic shifts. we formalized length attack, demystified it, and defended against it with LA(SER)$^3$. We found that, teaching the models "my longer-self = myself" provides a standalone semantic signal for more robust and powerful unsupervised representation learning.

---

[4] https://huggingface.co/datasets/sentence-transformers/embedding-training-data

## Limitations

We position that the focus of our work lies more in analyzing theoretical properties and inner-workings, and thus mostly focus on unsupervised contrastive learning settings due to compute constraints. However, we believe that with a better unsupervised checkpoint, further supervised fine-tuning will yield better results with robust patterns. We leave this line of exploration for future work. Further, we only focus on bi-encoder settings. In information retrieval, there are other methods involving using cross-encoders to conduct re-ranking, and sparse retrieval methods. Though we envision our method can be used as a plug-and-play module to many of these methods, it is hard to exhaust testing with every method. We thus experiment the plug-and-play setting with a few representative methods. We hope that future works could evaluate the effectiveness of our method combining with other lines of baseline methods such as cross-encoder re-ranking methods.

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

# A Results of STS-b

In this section, we present the results of STS-b test set (Table 5). As discussed in the main sections, we position that STS-b is not correlated with downstream semantic tasks performance (Reimers et al., 2016; Wang et al., 2021), and effectiveness of document-level representation encoders should be evaluated beyond this task. The inferior predictability of STS-B on downstream task performances have been attributed to length ranges (Abe et al., 2022). We hypothesize that, training with a large max sequence length increases the uncertainty of elongation hyperparameter $m$ of LA(SER)[3], resulting in a diverse length range, and less corresponding concrete examples at each length.

We show that, while out-performing SimCSE by a large margin on other downstream semantic tasks (Main Section, Table 1), our long sequence length poses a certain level of instability in converging, showing a small performance drop on shorter sentences (STS-b). The converging instability is further confirmed by training an extra LA(SER)[3] with `[cls]`-pooling, as `[cls]`-pooling is faster in converging - as it involves only optimizing one token. Notably, SimCSE also uses `[cls]`-pooling. Therefore, we roughly stay on-par with SimCSE on encoding shorter documents, while out-performing it by a large margin on other downstream tasks.

| Method | Max Seq. | sts-b |
|---|---|---|
| BERT-whitening | - | 68.19/71.34 |
| BERT-flow | 64 | 58.56/70.72 |
| SimCSE | 32 | 76.85 |
| LA(SER)[3]-mean | 256 | 75.61 |
| LA(SER)[3]-[cls] | 256 | 76.19 |

Table 5: STS-b test set results, compared with unsupervised sentence representation methods. SimCSE and LA(SER)[3] are trained on the same training$^{\text{wiki}}$. The two numbers of BERT-whitening and BERT-flow correspond to optimizing on NLI or target data (sts-b). Results are from the original works (Su et al., 2021; Li et al., 2020; Gao et al., 2021).