# OpenReview forum: "Length is a Curse and a Blessing for Document-level Semantics"
_EMNLP/2023/Conference — EMNLP 2023 Main_

### Official Review · Reviewer_ZBQ8 · 2023-08-02

**Soundness:** 3

**Excitement:**

4: Strong: This paper deepens the understanding of some phenomenon or lowers the barriers to an existing research direction.

**Paper Topic And Main Contributions:**

This paper discusses the vulnerability of contrastive learning (CL)-based models to length-induced semantic shift and proposes a length-agnostic self-reference framework for semantically robust sentence representation learning. The authors show that elongating a document intensifies the high intra-document similarity brought by CL and that isotropy promised by CL is highly dependent on the length range of text exposed in training. The proposed framework, LA(SER)^3, relies solely on the semantic signal provided by document length and achieves state-of-the-art unsupervised performance on a standard information retrieval benchmark.

**Reasons To Accept:**

The computation of sentence representations is a crucial aspect of natural language processing (NLP) tasks, with potential applications across various downstream tasks. Enhancing the resilience of encoding models is thus a significant undertaking in NLP research.

This paper presents a mathematical formulation of the elongation attack for contrastive learning. It describes the variations in semantics between the original text and the text that has undergone an elongation attack, and the shift in patterns of intra-document tokens following such an attack. This could potentially offer a comprehensible reference for future studies.

With the aid of the generalized mathematical formulation, this paper suggests two approaches that can effectively counter the elongation attack.


**Reasons To Reject:**

The writing and presentation of the paper require improvement, as the current version lacks distinctiveness as a novel contribution and may be challenging to differentiate from prior research [1].  The paper's introduction predominantly focuses on prior research. Similarly, Insufficient description of the proposed methods and the lack of formal representation of both methods make subsequent work difficult to fellow.

The proposed mathematical formulation lacks strict definition, as there exist various token forms that satisfy the $x_{dominat}$ definition. Consequently, this ambiguity could undermine the interpretability of the definition.

While the paper includes comprehensive quantitative experiments, the absence of analytical experiments for the elongation attack may impede the ability to ascertain the actual effectiveness of the proposed method.

[1]. Xiao C, Ye Z, Hudson G T, et al. Can Text Encoders be Deceived by Length Attack?[J]. 2023.

**Reproducibility:**

3: Could reproduce the results with some difficulty. The settings of parameters are underspecified or subjectively determined; the training/evaluation data are not widely available.

**Reviewer Confidence:**

3: Pretty sure, but there's a chance I missed something. Although I have a good feel for this area in general, I did not carefully check the paper's details, e.g., the math, experimental design, or novelty.

---

> ### Author Rebuttal · Authors · 2023-08-28
>
> We thank the reviewer for the insightful concerns.
>
> P1 (Writing and novelty): Thanks for pointing this out. In the final version, we will 1) add mathematical description of LA(SER)$^3$ self-reference and intra-reference in the method section to make it easier to follow. 2) improve the description of the novelty of this paper, including that we a) formally introduce the intra-document interaction change underlying elongation attacks through math derivation, b) conduct a systematic study of different elongation strategies (fixed-times longer or random-times longer) and their impacts on generalization, c) a novel intra-reference approach, which is shown to perform better than the self-reference approach.
>
> P2 (mathematical formulation): Indeed, there might be a set of dominant tokens that jointly decide the important aspects of the document-level semantics. Following the reviewer’s advice, we find that in order to introduce multiple dominant tokens, it might be most appropriate to introduce an “attention gain threshold” to decide whether a token belongs to “dominant tokens”. As our current math equations are defined to compute the attention gain of single tokens, if $X_{dominant}$ is now defined as multiple tokens, the new attention gain of dominant tokens will just be the sum of all the single dominant-token attention gains as defined in our current math equations. Notably, this adjustment does not change our derivation that elongation attacks make more attention lean towards dominant tokens than towards regular tokens.
>
> P3 (Analytical Experiments): Although we have presented two proof-of-concept experiments before proposing LA(SER)$^3$, we agree that we can benefit from more analytical experiments after the models have been trained with LA(SER)$^3$. In the final version, we will add the following experiments as also partly mentioned in replies to other reviewers:
> 1) We will present that LA(SER)$^3$ does make the models more robust to length attacks by showing a) LA(SER)$^3$ (BERT-base-uncased trained on wiki with unsupervised contrastive learning) perceives a more stable cosine similarity distribution between attacked text pairs (only shifted from the cosine similarity distribution between unattacked text pairs with a Jenson Shannon divergence of 0.17, compared to 0.39 JS divergence when encoded with its SimCSE counterpart). b) showing some examples wrongly retrieved by models like SimCSE because they are long and contain many misleading lexical overlaps but not necessarily a right match, but avoided by LA(SER)$^3$.
>
> 2) We will present a table to show the nDCG@10 per length range of documents retrieved by LA(SER)$^3$ models. We will show that LA(SER)$^3$ provides models with abilities to extrapolate accurate meanings in longer length ranges (by yielding a much higher nDCG@10 on longer texts), which the original datasets do not contain if we didn’t augment the data.
>
> We believe adding such analytical experiments will add on to the interpretability of LA(SER)$^3$ performance gain. We are happy to add more experiments if the reviewer has other suggestions!

---

### Official Review · Reviewer_wNeu · 2023-08-04

**Typos Grammar Style And Presentation Improvements:** N/A
**Soundness:** 4

**Excitement:**

3: Ambivalent: It has merits (e.g., it reports state-of-the-art results, the idea is nice), but there are key weaknesses (e.g., it describes incremental work), and it can significantly benefit from another round of revision. However, I won't object to accepting it if my co-reviewers champion it.

**Missing References:**

N/A

**Paper Topic And Main Contributions:**

This paper introduces LA(SER), a length-agnostic self-reference framework, to improve the robustness of contrastive learning-based models to document length. Experimental results demonstrate the effectiveness of LA(SER) in enhancing the performance of contrastive text encoders. The method leverages length as a semantic signal and conducts contrastive learning with elongated sentences to improve the robustness and isotropy of representations.

**Questions For The Authors:**

Please address the concerns in the "Reasons to Reject"

**Reasons To Accept:**

1. The paper provides an extensive analysis of the length generalizability of standard contrastive learning methods, addressing a significant yet overlooked research gap. The authors derive the theoretical foundations underlying length attacks and demonstrate that elongating a document intensifies the high intra-document similarity already present in contrastive learning. This analysis sheds light on the vulnerability of contrastive text encoders to length-induced semantic shifts, highlighting the importance of considering length in representation learning.

2. The paper proposes a simple yet universal framework called LA(SER) (Length-Agnostic Self-Reference for Semantically Robust Sentence Representation Learning) to bridge the unideal properties of contrastive learning. LA(SER) leverages the assumption that the elongated version of a sentence should still mean itself and should not become more or less similar to its pairs. By providing this simple signal, LA(SER) acts as an unsupervised contrastive learning method that enhances robustness to length attacks and improves encoding ability without sacrificing representational power. This framework offers a practical solution to address the length-based vulnerability of contrastive text encoders.

**Reasons To Reject:**

1. One weakness of the paper is that the evaluation of the proposed LA(SER) framework is primarily focused on unsupervised performance on the standard information retrieval benchmark. While this evaluation provides valuable insights into the effectiveness of LA(SER)3 in improving contrastive text encoders' robustness to length attacks, it would be beneficial to also evaluate the framework on other downstream tasks or benchmarks to assess its generalizability and applicability in different NLP domains.

2. The paper lacks a comprehensive comparison with existing methods that address the length vulnerability of contrastive learning. While the authors briefly mention related work on length preference of text encoders [4], there is no direct comparison with these methods or a discussion on how LA(SER)3 differs from or improves upon them. Including such a comparison would provide a clearer understanding of the unique contributions and advantages of LA(SER)3 in the context of existing approaches.

**Reproducibility:**

2: Would be hard pressed to reproduce the results. The contribution depends on data that are simply not available outside the author's institution or consortium; not enough details are provided.

**Reviewer Confidence:**

3: Pretty sure, but there's a chance I missed something. Although I have a good feel for this area in general, I did not carefully check the paper's details, e.g., the math, experimental design, or novelty.

---

> ### Author Rebuttal · Authors · 2023-08-28
>
> We thank the reviewer for the valuable concerns.
>
> W1 (Evaluation): Our answers to this question is three-fold:
> 1) As the main focus of our work on better dense retrievers, we would like to advocate the standardized evaluation of these models on tasks and real-life user cases that will benefit from the pipeline of “first pre-encode embeddings, then do inference with only similarity match”. Note that our evaluation is in line with some most established works in this field (e.g., Contriever [1], which is a widely-adopted unsupervised dense retrieval checkpoint; and also COCO-DR, which we compared to in our work as well).
>
> 2) We would also like to discuss that: it would make less sense to use representation models like ours to do other downstream tasks, as other downstream tasks (say, classification tasks) do more one-time inferences, and it will make more sense to directly finetune a vanilla model with some extra task-specific heads, instead of using a representation model to pre-compute the embeddings for multiple-times inferences (especially with similarity match as in retrieval tasks).
>
> 3) Apart from (document-level) dense retriever models, we also try to align with sentence-level representation models like SimCSE, InfoCSE, etc. As we discussed in the paper, the reason why we did not advocate evaluation on semantic textual similarity tasks is because previous works have found the performances on STS-b do not display strong correlations with downstream tasks [2,3] and this is because of its limited length range [4]. This becomes especially unsuitable because our work is on the problem of length, and needs evaluation from a wider length range. However, we also reported our sts-b performance in Appendix A for completeness, and discussed our performance.
> Hopefully these discussions clear the reviewer’s concern on evaluation!
>
> W2 (Compare with previous works): As the reviewer noted, we identified two works that are relevant [5,6]. 1) The BEIR paper ([5]) identified that models trained with dot-product and cosine similarity exhibit different length preferences, but didn’t propose a method to address this vulnerability, but only suggested training with either dot-product or cosine similarity based on specific user case. 2) The only paper that suggests an approach to combat this issue is from [6]. Our paper differs in terms of a full mathematical explanation for this phenomenon; a systematic study of different elongation times (fixed-times longer or random-times longer) and its relationship with generalization; and a novel intra-reference approach, which is shown to perform better than the self-reference approach (see Table 2).
>
> [1] Izacard, G., Caron, M., Hosseini, L., Riedel, S., Bojanowski, P., Joulin, A. and Grave, E., 2022. Unsupervised Dense Information Retrieval with Contrastive Learning. Transactions on Machine Learning Research.
>
> [2] Reimers, N., Beyer, P. and Gurevych, I., 2016, December. Task-oriented intrinsic evaluation of semantic textual similarity. In Proceedings of COLING 2016, the 26th International Conference on Computational Linguistics: Technical Papers (pp. 87-96).
>
> [3] Wang, K., Reimers, N. and Gurevych, I., 2021, November. TSDAE: Using Transformer-based Sequential Denoising Auto-Encoderfor Unsupervised Sentence Embedding Learning. In Findings of the Association for Computational Linguistics: EMNLP 2021 (pp. 671-688).
>
> [4] Abe, K., Yokoi, S., Kajiwara, T. and Inui, K., 2022, November. Why is sentence similarity benchmark not predictive of application-oriented task performance?. In Proceedings of the 3rd Workshop on Evaluation and Comparison of NLP Systems (pp. 70-87).
>
> [5] Thakur, N., Reimers, N., Rücklé, A., Srivastava, A. and Gurevych, I., 2021, August. BEIR: A Heterogeneous Benchmark for Zero-shot Evaluation of Information Retrieval Models. In Thirty-fifth Conference on Neural Information Processing Systems Datasets and Benchmarks Track (Round 2).
>
> [6] Xiao, C., Ye, Z., Hudson, G.T., Sun, Z., Blunsom, P. and Al Moubayed, N., 2023. Can Text Encoders be Deceived by Length Attack?.

---

### Official Review · Reviewer_oY3S · 2023-08-13

**Soundness:** 3

**Excitement:**

3: Ambivalent: It has merits (e.g., it reports state-of-the-art results, the idea is nice), but there are key weaknesses (e.g., it describes incremental work), and it can significantly benefit from another round of revision. However, I won't object to accepting it if my co-reviewers champion it.

**Paper Topic And Main Contributions:**

This paper studies the impact of sentence length on model performance. The authors find that in contrastive learning-based models, the meaning of the same sentence changes depending on the length of the sentence, suggesting that the existing contrastive frameworks suffer from the length vulnerability. The authors empirically found that models tend to rely on semantic signals from sentence length and that the isotropy of sentence embeddings is highly dependent on the length of the sentence used for training. Based on these two observations, the authors show that current contrastive frameworks are vulnerable to length and propose a length-agnostic contrastive learning framework to address this issue.

**Questions For The Authors:**

Please address the aforementioned weaknesses.

**Reasons To Accept:**

S1: The authors introduce examples of length preference in dense retrieval models and assume that contrastive learning based models are also vulnerable to length. This is assumed to be affecting the actual model performance, and two experiments were designed to prove it. This motivation is one of the strengths of this paper.

S2: The two experiments to demonstrate the length vulnerability are reasonably designed and I believe the results are actually sufficient to prove the length vulnerability issue of existing contrastive learning based models.

S3: The self-reference and intra-reference ideas to overcome the length vulnerability problem are effective.

**Reasons To Reject:**

W1: Motivation may come from sentences that are not actually used very often in real-life. Specifically, it is unlikely that the elongated sentences are commonly used in infromation retrieval or any other downstream task.  It is possible that the high similarity between the two elongated sentences is due to simple repetition. In the example in Figure 1, the (What is) token in the sentences S_a x 100 and S_b x100 is repeated, which may have led to the high similarity. In Figure 2, if S_a and S_b are subjected to the same m, it seems that the high similarity may be due to the many repetitions of the same token. I would like to hear the authors' opinions on this perspective.

W2: The proposed method utilizes the same loss as InfoCSE, and the only difference with InfoCSE is that it uses elongated sentences as positive pairs.

W3: As the authors mention as a limitation, they only experimented with BEIR. In Table 1, we can see that the experimental results are taken from InfoCSE's paper, and InfoCSE evaluated not only the IR task but also the NLU (GLUE Benchmark) performance. If there is an improvement in IR performance, but the performance in other tasks such as NLU decreases, it is difficult to see that the proposed method is meaningful. Therefore, experiments on various tasks are strongly required.

W4: Does the proposed method actually solve the length vulnerability problem? For example, it would be nice to visualize the distribution obtained by the model applied with the proposed solution, just like Figure 2.

W5: The authors set the default repeat size (m) for elongated sentences to 100, which seems too large to cover with a max sequence length of 256. How did you control this issue?

**Reproducibility:**

3: Could reproduce the results with some difficulty. The settings of parameters are underspecified or subjectively determined; the training/evaluation data are not widely available.

**Reviewer Confidence:**

4: Quite sure. I tried to check the important points carefully. It's unlikely, though conceivable, that I missed something that should affect my ratings.

---

> ### Author Rebuttal · Authors · 2023-08-28
>
> We thank the reviewer for the constructive feedback.
>
> W1 (The “What is” Word Overlap/Repetition): Our answers to this question are three-fold:
> 1) Exactly, there exists a “what if” as a word overlap in both sentences in the example. However, given a robust semantic encoder, we would expect conducting the copy elongation wouldn’t strengthen this “semantic alignment” that much (from 0.06 to 0.42). Using tf-idf on the other hand, will make the copy-concat elongation provide the same similarity as before.  In real life, if we search “what is NLP”, we wouldn’t want the search engine to return an article about teaching the grammar of “what is”, for instance, where there will probably be more “what is” in it.
>
> 2) Apart from this one example, we also conducted the same experiment on the corpus level (as shown in Figure 2 left), where there are a lot of sentence-pair examples that do not necessarily contain word overlapping, and the same trend holds.
>
> 3) From a dominant token perspective, “what is” is likely given very little attention after contrastive learning already (as we explained in Observation 2). Therefore, it shouldn’t be dominantly augmenting the semantics even if we repeat it. Thus, the augmented similarity in the example largely comes from repeating NLP and CV - the dominant tokens (they might contain some same properties in some embedding dimensions that make the cosine similarity much more alike after augmenting).
> Hope this discussion addresses the reviewer’s concern!
>
> W2 (Same loss): We use the standard InfoNCE loss as the contrastive loss, aligning with most papers in this research line (SimCSE, E-SimCSE, DiffCSE, InfoCSE, COCO-DR and more recent ones) and we do not benefit from an auxiliary network as in InfoCSE but still outperforms it in most IR tasks. We are one of the very few to systematically investigate the length generalization problems, and have combined the elongation training with our newly designed “intra-reference” method (Table 2). More importantly, it can be used as a plug-and-play module to any method that uses a contrastive loss.
>
> W3 (Evaluation): We would like to position that IR tasks provide the most effective way to test actually using the “frozen document embedding” to do inference only with similarity computation (Also importantly, IR provides a suite of tasks consisting of varied-length inputs). NLU tasks, on the other hand, require fully fine-tuning again and adding classification heads, which we position as not a “representation learning” problem. Were to solve NLU tasks, it would typically make more sense to finetune a vanilla model, instead of a model trained to encode representation. We also presented our sts-b results in Appendix A for completeness and also discussed why it is not good.
>
> W4 (Solve Length Vulnerability): Thanks for pointing this out! We will add a section to include more distribution plots in the camera-ready version given the chance. In short, yes, after our method, the misalignment in Figure 2 almost disappeared. Statistically, in an unsupervised setting trained on wiki with BERT-base-uncased, the Jenson Shannon Distance (JSD) between “Attacked v.s. Unattacked” goes from 0.39 (SimCSE) to 0.17 (LA(SER)$^3$, ours). In a supervised setting as in Figure 2, this number goes from around 0.6 to under 0.1 across all dataset settings we tried (e.g., QQP as in Figure 2, and MSMARCO, etc.). To compute the above values, we compute the cosine similarity distribution between unattacked sts-b sentence pairs (distribution 1), then compute the cosine similarity distribution  between original sentence 1s and sentence 2s attacked with random elongation (distribution 2), and compute the JS divergence between these 2 distributions. A lower number means that the semantics stay more stable with elongation attacks.
>
> W5 (elongation time m): The 100 times the reviewer saw is from the proof-of-concept experiment. In training, we explain the selection of m from line 484 to line 490, and also in ablation analysis. We sample an input-dependent m. If an input is 50-token long, we sample m from [1,5] to prevent it from exceeding 256. If it’s 10-token long, we sample m from [1,25], making sure it’s shorter than 256 tokens.

---

### Official Review · Reviewer_omiY · 2023-08-14

**Soundness:** 5

**Excitement:**

4: Strong: This paper deepens the understanding of some phenomenon or lowers the barriers to an existing research direction.

**Paper Topic And Main Contributions:**

The paper tackles an interesting and important problem of length bias in contrastive learning methods. The paper demonstrates interesting insight such as the effect of length based attacks on document similarity, further exemplified by contrastive learning. The work also shows that length also affects the geometry of the embedding space as the isotropy guarantees are not length agnostic. The authors also empirically demonstrate this and propose a length agnostic approach to increase robustness to length attacks through two augmentation approaches. The results in standard benchmarks seem encouraging in most cases.

**Questions For The Authors:**

The authors must discuss on how they handle the context length issue in encoders for very large elongated sequences. Do they employ linear attention mechanisms like reformer ? If not can authors report results using these encoders as they are a fairer evaluation for elongated sequences without truncation imposed to fit in memory.

**Reasons To Accept:**

1. The paper studies an important problem of length bias in dense retrievers. The proposed approach to tackle the same is intuitive and simple.

2. the effect of length attacks exemplified by contrastive learning is explained well along with supporting experiments. The insight that the token interaction also change in a contrastive framework is interesting. the study of effects of length interplay with contrastive training on the geometry of the embedding space is interesting too.

3. The experimental results seem convincing though the proposed approach does not give uniform gains on all datasets, overall it seems to perform well compared to existing approaches and relatively more robust to length based attacks.

**Reasons To Reject:**

1. The proposed augmentation approaches based only on elongation are not convincing as depending on training sample size the model may overfit to this particular augmentation resulting in catastrophic forgetting. i believe a detailed study on whether the model fine tuned with LASER forgets other semantic characteristics learned is necessary. Also to evaluate robustness  different encoders must be evaluated.

2. I'm not entirely convinced of the positional invariance argument. While set based applications require position invariance and self attention is by definition invariant unless added with positional embeddings other retrieval applications are position dependent. For example extractive summarization depends on position as terms appearing earlier in document are more relevant. And attention coupled with positional embeddings help surface this.  Authors must spend more effort in empirically showing this argument before making the assumption of positional embeddings having little impact.

3. More detailed discussion of results would be appreciated. Also statistical significance tests would help determine the superiority of proposed approach in cases where the margin of gain is low.

**Reproducibility:**

4: Could mostly reproduce the results, but there may be some variation because of sample variance or minor variations in their interpretation of the protocol or method.

**Reviewer Confidence:**

5: Positive that my evaluation is correct. I read the paper very carefully and I am very familiar with related work.

---

> ### Author Rebuttal · Authors · 2023-08-28
>
> We thank the reviewer for the valuable comments!
>
> P1 (Positional invariance): To address this concern, we have now conducted an extra group of experiments to present the sensitivity change of language models towards positions before and after contrastive learning.
> We take 4 models (MiniLM and mpnet respectively before and after contrastive learning on [1]). We take sentence pairs from sts-b, compute their perceived cosine similarity (distribution 1), and then randomly shuffle the word orders of all sentence 1s in the sentence pairs, and compute their cosine similarity with sentence 2s again (distribution 2), and measure the divergence of these 2 distributions with Jenson Shannon Divergence.
> We find that the JS divergence yielded by MiniLM has gone from 0.766 (vanilla) to 0.258 (after contrastive learning). And the same for Mpnet goes from 0.819 (vanilla) to 0.302 (after contrastive learning). Even though 0.258 and 0.302 are still not too close to 0, we see that the distributions almost fully overlap with each other in cases for both models  after contrastive learning. This finding has shown that contrastive learning has largely removed the contribution/sensitivity of positions, even in the most extreme case (with random shuffled word orders). This has made contrastively-learned models acting more like bag-of-words models, aligning with what was previously found in vision-language models as well [2].
> Moreover, MiniLM uses absolute positional embeddings while mpnet applies further relative positional embeddings. We believe that the positional insensitivity pattern holds for both models can also partly contribute to solving the reviewer’s concern on experimenting with all kinds of encoders (with different positional embedding methods).
>
> P2 (Catastrophic forgetting and overfitting): We partly discuss this issue in ablation analysis through the lens of elongation times m. We find that using a fixed-times elongation (say 2 times) might induce this kind of pattern overfitting, and probably does not teach the model to extrapolate meanings in unseen length ranges, but only teaches the model that 1=2. And using a random-times elongation is more extrapolative, as it’s essentially teaching the model to know that “Thanks for your valuable comments” means the same thing in position [1,5] just as what it means in position [5m+1,5m+5], for instance. We envision that the randomness in augmentation is robustness-preserving. Moreover, as the reviewer suggested, we aim to include some linguistic characteristics tests that could be done via probing the encoded embeddings in the final version given the chance.
>
> P3 (Discussions on results where margin of gain is low): In general, we find that our performance gain is more pronounced when the length range of the dataset is large. For instance, on BERT-base experiments, nfcorpus (doc. avg. length 232.26, nDCG@10 performance 0.1048 (SimCSE) -> 0.1919 (Ours)), scifact (doc. avg. length 213.63, nDCG@10 performance 0.2492 (SimCSE) -> 0.4317 (Ours)), Arguana (doc. avg. length 166.80, performance 0.2796 (SimCSE) -> 0.4227 (Ours)). On the other hand, our performance gain is low when documents are shorter, such DBPedia (avg. length 49.68) and Quora (avg. length 11.44). Results on MiniLM-L6 are a bit more nuanced, which we think is because the linguistic knowledge has been more unstable after every second layer of the model is taken (from 12 layers in MiniLM-L12 to 6 layers). We believe the ability to extrapolate meanings for longer text is beneficial in use cases where the domain and task-specific training set has limited length range but the model needs to face unexpected text range in inference time.
>
> P4 (Handle Context Length Issue for very large elongated sequences): We explain how we sample input-dependent elongation times in the paper (line 484 to line 490). In short, we do not make the elongated sequences too long. If our max sequence length is 256 in training, we randomly sample an elongation times from 1-25 times if an input is 10-token long excluding [cls], and 1-5 times if it’s 50-token long. This means that we do not elongate sequences that are originally longer than 128. This has ensured a diversity for our elongation times m, and teaches the model to extrapolate even to a longer length (with the learned pattern of 1= m). With this extrapolation, we do not need to train the models explicitly on say 1000-token inputs but the model will still be better than the baselines on dealing with 1000-token inputs.
> Therefore, we speculate that training with linear attention models that might be more beneficial when explicitly trained with longer texts, might not be as rewarding as the extrapolative ability itself. In line with our findings on BERT, we conjecture that it takes a larger batch size for LA(SER)$^3$ to show its advantages, when training on longer sequence lengths with linear attention models, and it’ll potentially be more beneficial to incorporate gradient caching for larger batch sizes, when training them. Due to compute constraints, we struggle to finish this part of experiments during rebuttal and will release the results in the final version for completeness.
>
> [1] https://huggingface.co/datasets/sentence-transformers/embedding-training-data
>
> [2] Yuksekgonul, M., Bianchi, F., Kalluri, P., Jurafsky, D. and Zou, J., 2022, September. When and Why Vision-Language Models Behave like Bags-Of-Words, and What to Do About It?. In The Eleventh International Conference on Learning Representations.

---

### Meta-Review · Area_Chair_rXPw · 2023-09-18

**Recommendation:** 4

**Metareview:**

In this work the authors extensively analyze the effect of sentence/document length in contrastive training of sentence/document encoders. They show that contrastive training, recently ubiqitous in training sentence encoders, introduces a substantial length bias. The authors propose to reduce the length bias of the resulting embedding space by means of training data augmentation, i.e., synthetic elongation of training sentences. The effectiveness of the proposed approach is evaluated on the BEIR benchmark, which encompasses 14 different IR tasks with "documents" of varying lengths (sentences, tweets, news stories, ...).

The reviewers are generally appreciative of the analysis of length biases in representations of encoders trained with contrastive objectives and the proposed augmentation-based remedy. Concerns raised w.r.t. shown positional invariance (i.e., that length matters more than the position of the content in the text) and downstream NLU evaluation have been, in my opinion, convincingly addressed by the authors, both by means of reporting additional experiments and acceptable argumentation (e.g., that one would fine-tune vanilla LMs rather than sentence/text encoders for supervised NLU tasks). Some concerns, however, have not been fully addressed: question of direct comparison against the approach of Xiao et al. (ICLR 23), and that of performance on STS (unsupervised, similarity-based).

The work of Xiao et al in principle challenges the novelty of the contributions of this work -- as they also demonstrate that contrastive learning leads to overfitting to lengths observed in training data. I believe, however, that this work can be seen as concurrent/contemporary: it would still require that the authors compare against this work in the revised version of this work. The argument that authors offer against STS is that of limited length variance in the dataset, but I did not find this to be particularly convincing. STS results seem to show that LA(SER)^3, i.e., the augmentation strategy, seems to hurt the performance of short sentences.

Overall, however, the contributions of this work outweigh the remaining concerns (which, however, I still believe should be remedied in the revised version of the paper)

---

### Decision · Program_Chairs · 2023-10-07

**Decision:**

Accept-Main

**Comment:**

In this work the authors extensively analyze the effect of sentence/document length in contrastive training of sentence/document encoders. They show that contrastive training, recently ubiqitous in training sentence encoders, introduces a substantial length bias. The authors propose to reduce the length bias of the resulting embedding space by means of training data augmentation, i.e., synthetic elongation of training sentences. The effectiveness of the proposed approach is evaluated on the BEIR benchmark, which encompasses 14 different IR tasks with "documents" of varying lengths (sentences, tweets, news stories, ...).

The reviewers are generally appreciative of the analysis of length biases in representations of encoders trained with contrastive objectives and the proposed augmentation-based remedy. Concerns raised w.r.t. shown positional invariance (i.e., that length matters more than the position of the content in the text) and downstream NLU evaluation have been, in my opinion, convincingly addressed by the authors, both by means of reporting additional experiments and acceptable argumentation (e.g., that one would fine-tune vanilla LMs rather than sentence/text encoders for supervised NLU tasks). Some concerns, however, have not been fully addressed: question of direct comparison against the approach of Xiao et al. (ICLR 23), and that of performance on STS (unsupervised, similarity-based).

The work of Xiao et al in principle challenges the novelty of the contributions of this work -- as they also demonstrate that contrastive learning leads to overfitting to lengths observed in training data. I believe, however, that this work can be seen as concurrent/contemporary: it would still require that the authors compare against this work in the revised version of this work. The argument that authors offer against STS is that of limited length variance in the dataset, but I did not find this to be particularly convincing. STS results seem to show that LA(SER)^3, i.e., the augmentation strategy, seems to hurt the performance of short sentences.

Overall, however, the contributions of this work outweigh the remaining concerns (which, however, I still believe should be remedied in the revised version of the paper)